# Time-Domain Dynamic Characteristics Analysis and Experimental Research of Tri-Stable Piezoelectric Energy Harvester

**DOI:** 10.3390/mi12091045

**Published:** 2021-08-29

**Authors:** Xuhui Zhang, Luyang Chen, Xiaoyu Chen, Fulin Zhu, Yan Guo

**Affiliations:** 1College of Mechanical Engineering, Xi’an University of Science and Technology, Xi’an 710054, China; chenluyang@stu.xust.edu.cn (L.C.); 19105016004@stu.xust.edu.cn (X.C.); 20205224060@stu.xust.edu.cn (F.Z.); g1014901143@163.com (Y.G.); 2Shaanxi Key Laboratory of Mine Electromechanical Equipment Intelligent Monitoring, Xi’an University of Science and Technology, Xi’an 710054, China; 3College of Engineering, Zunyi Normal College, Zunyi 563006, China

**Keywords:** linear-arch beam, tri-stable piezoelectric energy harvester, nonlinear magnetic force, dynamic modeling

## Abstract

In order to explore the dynamic characteristics of the linear-arch beam tri-stable piezoelectric energy harvester (TPEH), a magnetic force model was established by the magnetic dipole method, and the linear-arch composite beam nonlinear restoring force model was obtained through experiments. Based on the Euler–Bernoulli beam theory, a system dynamic model is established, and the influence of the horizontal distance, vertical distance and excitation acceleration of magnets on the dynamic characteristics of the system is simulated and analyzed. Moreover, the correctness of the theoretical results is verified by experiments. The results show that the system can be mono-stable, bi-stable and tri-stable by adjusting the horizontal or vertical spacing of the magnets under proper excitation. The potential well of the system in the tri-stable state is shallow, and it is easier to achieve a large-amplitude response. Increasing the excitation level is beneficial for the large-amplitude response of the system. This study provides theoretical guidance for the design of linear-arch beam TPEH.

## 1. Introduction

In recent years, the application of wireless sensing technology in equipment monitoring, environmental monitoring and other fields has attracted more and more researchers’ interest [1,2,3]. However, the battery life of wireless sensor nodes restricts its development, and the use of chemical batteries brings about high cost, environmental pollution and limited life span [4]. Obtaining energy from the environment has the potential to solve this problem and to address the power supply issue. Vibration energy has great advantages due to its ubiquity and high energy density. The piezoelectric vibration energy harvester has the advantages of a simple structure, compact size and easy integration and has broad application prospects [5].

In order to realize the collection and conversion of vibration energy, the piezoelectric cantilever beam has a simple structure and is prone to large strain after being excited. Therefore, domestic and foreign researchers have carried out a lot of research work on the cantilever beam piezoelectric energy harvester [6,7,8,9]. The traditional linear piezoelectric energy harvester can only work in a limited bandwidth where the environmental excitation is very close to its resonance frequency. When the environmental vibration frequency is far away from the resonance frequency of the device, the energy that can be captured is significantly reduced. This makes the energy harvester inefficient in practical applications, because most environmental vibrations occur randomly in a wide frequency range. Therefore, it is urgent to find a way to broaden the effective working frequency bandwidth to improve the performance of the vibration energy harvester [10]. The method of widening the working frequency bandwidth can be divided into two methods: linear frequency extension and nonlinear frequency extension [11]. The linear methods are array method [12], L-shaped beams [13], multi-degree-of-freedom beams [14] and nested structures [15]. Although these methods can effectively broaden the frequency band, the operating band width is still narrow for a single cantilever beam in these structures, and the system structure size is large. The efficiency of energy capture per unit volume is not high. Compared with the linear frequency extension method, the nonlinear technology can broaden the working frequency band of a single cantilever beam, thereby improving the power generation efficiency. Rezaei et al. [16] installed a spring on the free end of a linear cantilever beam. The research results show that the introduction of a nonlinear spring effectively broadens the resonance frequency band of the energy harvester. Liu et al. [17] designed an energy harvester with mechanical stoppers. The experimental results show that when the limiters are installed on both sides of the cantilever beam, the working frequency band can be broadened to 18 Hz.

In various nonlinear frequency extension structures and methods, the introduction of nonlinear magnetic force enables the energy harvester to work under nonlinear conditions of bi-stable, tri-stable and even more stable states [18,19,20,21]. Yao [22] proposed an L-beam bi-stable piezoelectric energy harvester. The research results show that the introduction of the end mass and nonlinear magnetic force increases the voltage and expands the effective frequency bandwidth. Wang [23] conducted research on the nonlinear magnetic force modeling of the tri-stable piezoelectric energy harvester, proposed an improved magnetic force model to describe the nonlinear magnetic force, and explored the influence of the magnet spacing on the magnetic force. Zhu [24] and Zhou [25] compared and analyzed the performance of the tri-stable piezoelectric energy harvester and the bi-stable piezoelectric energy harvester from the perspective of the frequency domain response characteristics of the energy harvester. The research results show that the tri-stable piezoelectric energy harvester has a shallower potential well, a wider trapping energy band and a higher output. Zhang [26] et al. designed a combined beam type tri-stable piezoelectric energy harvester, established a distributed parameter model based on the Hamiltonian principle, and analyzed the influence of the horizontal distance of the magnet and the excitation acceleration on the amplitude-frequency response characteristics from the perspective of the frequency domain. In order to improve the energy trapping efficiency, Li [27] proposed a tri-stable piezoelectric energy harvester with a trapezoidal well. Through the asymmetrical arrangement of the external magnet of the cantilever beam, the research results show that this design makes the large response easier to be triggered. Wang [28] derives the distributed parameter model of the energy harvester on the basis of considering the geometric nonlinearity of the cantilever beam (GNL) and the gravitational effect (GE). Theoretical and experimental studies have shown that the tri-stable energy harvester considering GNL and GE has an asymmetric potential well, which can improve the output performance. Unlike the conventional tri-stable energy harvester structure, Sun [29] also proposed a new type of tri-stable piezoelectric energy harvester with only a ring magnet outside, which can present different dynamic characteristics by adjusting the distance of the magnet. Most of the abovementioned energy harvesters that introduce a nonlinear magnetic field use classic straight beam, which cannot effectively collect multidirectional vibration energy in the environment, which limits the application of energy harvester in actual environments.

Aiming at the power supply problem of wireless monitoring nodes in coal mines and considering the characteristics of multidirectional and low-frequency excavation excitation, Zhang [30] proposed a new type of multifield coupled multidirectional piezoelectric energy harvester. As shown in Figure 1, the device can adapt to the mining environment and effectively collect vibration energy in different directions. Based on this, this paper designs a linear-arch beam TPEH. Under u-direction excitation conditions, a linear-arch beam TPEH dynamic model is established, and the influence of the horizontal distance, vertical distance and excitation acceleration of the magnet on the dynamic response characteristics is analyzed by means of numerical simulation. An experimental platform was built to verify the correctness of the theoretical analysis, the research provided theoretical guidance for the design of the linear-arch beam TPEH. The structure of this article is arranged as follows: Section 2 provides a schematic diagram of the structure of the TPEH in this study and then introduces the establishment of the nonlinear restoring force model and the magnetic force model, and the dynamic model is obtained. In Section 3, the system potential energy under different magnet distances and the dynamic performance under different magnetic distances and excitation accelerations are analyzed through simulation. In Section 4, experiments were performed to verify the theoretical results. Finally, in Section 5, the main findings and conclusions are presented and summarized.

## 2. Structure and Theoretical Model of a TPEH

### 2.1. Structure of the TPEH

Figure 2 shows the principle diagram of the structure of the TPEH. The system consists of a linear-arch composite beam, piezoelectric materials and three magnets. The linear-arch composite beam is pasted with flexible piezoelectric material PVDF; magnet A is fixed at the end of the cantilever beam; magnets B and C are symmetrically arranged on both sides of the x-axis; the horizontal distance between magnet A and B or C is d; and the vertical distance between magnets B and C is $2d_{g}$. In the figure, the length of the composite beam in the x-axis direction is L; the width of the composite beam is w; the thickness is $h_{s}$; the length of the linear beam is $L_{1}$; and the radius and chord length of the arched part are r and 2r.The width of the PVDF pasted on the composite beam is the same as that of the composite beam; the length is $L_{2}$; and the thickness is $h_{p}$.

### 2.2. Theoretical Modeling

#### 2.2.1. Nonlinear Restoring Force of Linear-Arch Beam

Unlike the linear restoring force of the typical straight beam, the restoring force is nonlinear due to the arched structure in the linear-arched beam. The YLK-10 dynamometer is used to measure the restoring force of the composite beam structure in the z-axis direction, the average of multiple measurements is taken, and the curve-fitting method is used to obtain multiple expressions of the nonlinear restoring force
(1)$$F_{r}=s_{3}w{\left(L,t\right)}^{3}+s_{2}w{\left(L,t\right)}^{2}+s_{1}w\left(L,t\right)$$
where $s_{1},\;s_{2}$ and $s_{3}$ are polynomial coefficients, and w(L,t) is the displacement of the beam along the z-axis at time t.

Figure 3 shows the experimental measurement and curve-fitting results of the linear-arch beam’s nonlinear restoring force. After fitting, the polynomial coefficients $s_{1}=-14\;N/m$, $s_{2}=254.586\;N/m^{2}$, $s_{3}=-56,681.2\;N/m^{3}$. It can be observed from the figure that the restoring force of the beam is a curve due to the existence of the arched part. Taking w(L,t)=0 as the equilibrium position of the cantilever beam, the restoring forces of the beam are asymmetric. This is because in the process of beam deformation, when the curvature of the arched part becomes larger, the force required is smaller than when the curvature becomes smaller. In the measurement results of the restoring force of the composite beam, the restoring force on both sides of the balance point is not completely symmetrical.

To determine the vibration displacement w(x,t) of the beam, the Rayleigh-Ritz method is used to expand the vibration displacement of the composite beam as
(2)$$w\left(x,t\right)=\overset{n}{\underset{i=1}{\sum }}\varphi_{i}\left(x\right)q_{i}\left(t\right)$$
where i is the order of the vibration mode of the composite beam, $\varphi_{i}\left(x\right)$ represents the *i*-th modal function of the composite beam, and $q_{i}\left(t\right)$ represents the *i*-th generalized modal coordinate. For the linear-arch beam in this paper, one end is clamped and fixed on the base, and the other end is free. The allowable function can be expressed as [31]
(3)$$\varphi_{i}\left(x\right)=1-\cos\left[\frac{\left(2i-1\right)\pi x}{2L}\right]$$

Since the excitation of piezoelectric energy harvester is mainly low frequency, the first-order modal bending vibration of the beam plays a leading role; therefore, this paper only considers the first-order mode of the beam.

#### 2.2.2. Nonlinear Magnetic Force Modeling

In order to accurately analyze the vibration characteristics of a piezoelectric cantilever beam, it is necessary to determine the magnitude of the nonlinear magnetic force at its end. The geometric relationship between magnets A, B and C is shown in Figure 4. This paper uses a magnetic dipole model to describe the nonlinear magnetic force.

The magnetic flux density generated by magnet B at magnet A is
(4)$$B_{BA}=-\frac{\mu_{0}}{4\pi }\nabla \frac{M_{B}\vec{r_{BA}}}{{\left\|\vec{r_{BA}}\right\|}_{2}^{3}}$$
where $\mu_{0}$ is the vacuum permeability, ∇ is the vector gradient, $\vec{r_{BA}}$ is the direction vector from magnet B to A, and $M_{B}$ is the magnetic moment of the magnetic dipole B. Then, the potential energy generated by the magnet B at the magnet A is
(5)$$U_{BA}=-B_{BA}\cdot M_{A}$$

Compared with the composite beam size, the magnet size is smaller; therefore, $l_{a}sin\alpha \ll w\left(L,t\right)$, so ∆x≅0 [32], so
(6)$$r_{BA}=-d\vec{i}+\left[w\left(L,t\right)-d_{g}\right]\vec{j}\;M_{A}=m_{A}V_{A}cos\alpha \vec{i}+m_{A}V_{A}sin\alpha \vec{j}\;M_{B}=-m_{B}V_{B}\vec{i}$$
where $\vec{i}$ and $\vec{j}$ are the unit vectors in the x and z axis directions, respectively; $m_{A}$ and $m_{B}$ represent the magnetization of magnets A and B, respectively; and $V_{A}$ and $V_{B}$ represent the volumes of magnets A and B. Since $\alpha =\arctan\left[w^{\prime }\left(L,t\right)\right]$, we have
(7)$$cos\alpha =\frac{1}{\sqrt{{\left[w^{\prime }\left(L,t\right)\right]}^{2}+1}},sin\alpha =\frac{w^{\prime }\left(L,t\right)}{\sqrt{{\left[w^{\prime }\left(L,t\right)\right]}^{2}+1}}$$

Substituting Formulaes (4), (6) and (7) into Formula (5)
(8)$$U_{BA}=\frac{\mu_{0}m_{A}V_{A}m_{B}V_{B}*\left\{-{\left[w\left(L,t\right)-d_{g}\right]}^{2}+2d^{2}-3d\left[w\left(L,t\right)-d_{g}\right]w^{\prime }\left(L,t\right)\right\}}{4\pi \sqrt{{\left[w^{\prime }\left(L,t\right)\right]}^{2}+1}*{\left\{d^{2}+{\left[w\left(L,t\right)-d_{g}\right]}^{2}\right\}}^{5/2}}$$
in the same way, the potential energy generated by magnet C at magnet A is as follows:(9)$$U_{CA}=\frac{\mu_{0}m_{A}V_{A}m_{C}V_{C}*\left\{-{\left[w\left(L,t\right)+d_{g}\right]}^{2}+2d^{2}-3d\left[w\left(L,t\right)+d_{g}\right]w^{\prime }\left(L,t\right)\right\}}{4\pi \sqrt{{\left[w^{\prime }\left(L,t\right)\right]}^{2}+1}*{\left\{d^{2}+{\left[w\left(L,t\right)+d_{g}\right]}^{2}\right\}}^{5/2}}$$

The potential energy generated by magnets B and C at magnet A at the end of the beam is
(10)$$U_{m}=U_{BA}+U_{CA}$$

The magnetic force on magnet A is
(11)$$F_{m}=\frac{\partial U_{m}}{\partial q\left(t\right)}$$

Using polynomial fitting to simplify the magnetic force to a polynomial about the displacement w(L,t), we can obtain
(12)$$F_{M}=K_{1}w{\left(L,t\right)}^{5}+K_{2}w{\left(L,t\right)}^{3}+K_{3}w\left(L,t\right)$$
where $K_{1}$, $K_{2}$ and $K_{3}$ are obtained by curve fitting.

#### 2.2.3. Dynamic Modeling

In this paper, Lagrange’s equation is used to establish the motion equation of the linear-arch beam
(13)$$L_{a}\left(x,t\right)=T_{S}+T_{P}+T_{M}+W_{P}-U_{r}-U_{M}$$
where $T_{S}$ is the kinetic energy of the metal base layer, $T_{P}$ is the kinetic energy of the piezoelectric layer, $T_{M}$ is the kinetic energy of the end magnet, $W_{P}$ is the electrical energy of the piezoelectric layer, and $U_{r}$ is the potential energy of the linear-arch piezoelectric cantilever beam.
(14)$$T_{S}=\frac{1}{2}\rho_{S}A_{S}\int_{0}^{L}{\left(\dot{w}\left(x,t\right)+\dot{z}\left(t\right)\right)}^{2}dx$$
(15)$$T_{P}=\frac{1}{2}\rho_{P}A_{P}\int_{0}^{L}{\left(\dot{w}\left(x,t\right)+\dot{z}\left(t\right)\right)}^{2}dx$$
(16)$$T_{M}=\frac{1}{2}M_{t}{\left({\left[\dot{w}\left(x,t\right)\right]}_{x=l}+\dot{z}\left(t\right)\right)}^{2}+\frac{1}{2}I_{t}{\left({\left[\frac{\partial^{2}w\left(x,t\right)}{\partial t\partial x}\right]}_{x=l}\right)}^{2}$$
(17)$$W_{P}=\frac{1}{2}\int_{V_{0P}}E_{3}D_{3}dV_{P}=\frac{1}{4}e_{31}b\left(h_{S}+h_{P}\right)v\left(t\right){\left[\frac{\partial w\left(x,t\right)}{\partial x}\right]}_{x=l}+\frac{1}{2}C_{P}v^{2}\left(t\right)$$
(18)$$U_{r}=\int F_{r}\;dq\left(t\right)$$
where “˙” means derivation with respect to t and “${}^{\prime }$” means derivation with respect to x. $A_{s}$ is the cross-sectional area of the metal base layer, $A_{P}$ is the cross-sectional area of the piezoelectric layer, w(x,t) is the vibration displacement of the composite beam, z(t) is the vibration displacement of the base, $M_{t}$ is the mass of the magnet, and $I_{t}$ is the moment of inertia of the magnet.

According to the Euler–Bernoulli beam theory and Kirchhoff’s law, the system dynamics equations of linear-arch composite beams can be obtained
(19)$$M\ddot{q\left(t\right)}+C\dot{q\left(t\right)}+F_{r}-\theta v\left(t\right)+F_{M}=-H\ddot{Z\left(t\right)}$$
(20)$$\theta \dot{q\left(t\right)}+C_{P}\dot{v\left(t\right)}+\frac{v\left(t\right)}{R}=0$$
where
(21)$$M=\left(\rho_{S}A_{S}+\rho_{P}A_{P}\right)\int_{0}^{L}{\left(\varphi \left(x\right)\right)}^{2}dx+M_{t}{\left(\varphi \left(L\right)\right)}^{2}+I_{t}{\left(\varphi^{\prime }\left(L\right)\right)}^{2}$$
(22)$$\theta =\frac{1}{2}e_{31}b\left(h_{S}+h_{P}\right)\varphi^{\prime }\left(L\right)$$
(23)$$H=\left(\rho_{S}A_{S}+\rho_{P}A_{P}\right)\int_{0}^{L}\varphi \left(x\right)dx+M_{t}\varphi \left(L\right)$$

The following nondimensionalized parameters are introduced [33]:(24)$$x\left(\tau \right)=\frac{q\left(t\right)}{l},\;\mu \left(\tau \right)=\frac{v\left(t\right)}{e},\;\tau =\omega_{0}t,\;e=\frac{L\theta }{C_{p}}$$
where l is the length coefficient; e is the voltage coefficient; τ is the time coefficient; and $\omega_{0}=\sqrt{\frac{K}{M}}$.

Incorporating Equation (24) into Equations (19) and (20), the dimensionless dynamic equation of the system is
(25)$$\ddot{x}+2\zeta \dot{x}-\vartheta \mu +\delta \left(F_{m}+F_{r}\right)=-\beta sin\left(\Omega \tau \right)$$
(26)$$\dot{x}+\dot{\mu }+\alpha \mu =0$$
where $\zeta =\frac{C}{2\omega_{0}M}$, $\vartheta =\frac{\theta^{2}}{KC_{P}}$,$\;\alpha =\frac{1}{\omega_{0}RC_{P}}$,$\;\delta =\frac{1}{KL}$, $\beta =\frac{HA}{KL}$, $\Omega =\frac{\omega }{\omega_{0}}$.

Let,$\;x_{1}=x$, $x_{2}=\dot{x}$,$\;x_{3}=u.$ We can obtain the state space equation:(27)$$\{\begin{array}{c} \dot{x_{1}}=x_{2}\\ \dot{x_{2}}=-2\xi x_{2}-\vartheta x_{3}-\delta \left(F_{m}+F_{r}\right)-\beta sin\left(\Omega \tau \right)\\ \dot{x_{3}}=-x_{2}-\alpha x_{3} \end{array}$$

## 3. Dynamic Analysis of TPEH

### 3.1. Analysis of System Potential Energy

Table 1 shows the structural parameters of the beam. The total potential energy P of the system can be expressed as
(28)$$P=U_{m}-U_{b}$$

According to Formula (28), the system parameters of horizontal magnetic distance d and vertical magnetic distance $d_{g}$ play a decisive role in the potential energy of the system. This section uses simulation to analyze the influence of parameters on the potential energy of the system.

Figure 5a shows the simulation results of the potential energy of the TPEH when $d_{g}=8\;\mathrm{mm}$ and d is 7, 11, 13, 15 and 22 mm, respectively. It can be observed that with the decrease in d, the potential-energy curve changes from a single potential well to a triple potential well, and the depth of the potential well gradually increases, because the magnetic force gradually increases with the gradual decrease in *d*.

Figure 5b shows the potential energy simulation results of the TPEH, when d=16 mm, and $d_{g}$ is 0, 6, 8, 12 and 25 mm. With the increase in $d_{g}$, the system goes through three motion states of bi-stable, tri-stable and mono-stable in sequence. When $d_{g}=0\;\mathrm{mm}$, the magnets B and C coincide, and the potential energy curve of the system has two potential wells. With the gradual increase in *d_g_*, the potential energy curve of the system gradually changes from two potential wells to three potential wells. With the increase in $d_{g}$, the depth of the potential well in the middle of the system becomes deeper, and the width increases; while the depth of the potential wells on both sides becomes shallower, the width is reduced. As dg continues to increase to 25 mm, the force between magnets A, B, and C is very small and almost zero. The system has only a single potential well, which appears as a mono-stable system. It can be observed from Figure 4a,b that especially when there are three potential wells in the potential energy curve, the two external potential wells are asymmetrical: one is high, and the other is low. This is due to the asymmetric potential well caused by the asymmetry of the restoring force.

### 3.2. Dynamic Analysis

Combining the results of the potential energy analysis, it can be observed that the magnet spacing has a significant effect on the system dynamics. This section explores the influence of the magnet horizontal spacing d, vertical spacing $d_{g}$ and excitation acceleration a on the system dynamics characteristics.

#### 3.2.1. The Influence of Horizontal Magnetic Distance d

When $A=12\;m/s^{2}$,$d_{g}=8\;\mathrm{mm}$, Ω=0.5, changing the horizontal magnetic distance d, the displacement–velocity phase diagram and time-displacement history diagram of the TPEH are shown in Figure 6.

As shown in Figure 6a, when the horizontal distance between the magnets is d=22 mm, the force between the magnets is small and has almost no effect on the beam. Therefore, the piezoelectric energy harvester exhibits mono-stable motion characteristics. With the decrease in d, the force between the magnets increases, and the influence of magnetic force on the motion characteristics of the energy harvester gradually appears, and the system changes from mono-stable to tri-stable. When d=15 mm, the system can cross the potential barrier, the beam moves back and forth between the three equilibrium points. From Figure 6b, it can be observed that the system exhibits tri-stable motion characteristics, reaching a large-amplitude response; the dimensionless displacement is 0.45; and the response displacement reaches 18 mm. When the horizontal distance between the magnets is reduced to 13 mm, due to the large force between the magnets, the beam struggles to get rid of the restraint of the magnetic force. As shown in Figure 6c, the system exhibits mono-stable motion characteristics. The end of the beam makes a small periodical movement near the center balance point, and the system response displacement and output voltage are very small.

#### 3.2.2. The Influence of Vertical Magnetic Distance $d_{g}$

When $A=12\;m/s^{2}$, d=16mm, Ω=0.5, changing the magnet vertical distance $d_{g}$, the displacement-velocity phase diagram and time history diagram of the TPEH are shown in Figure 7.

When the vertical distance between the magnets B and C is small, the system potential energy curve has two potential wells. When the excitation level is low, the system cannot cross the potential barrier and can only move in the well; therefore, the system cannot exhibit bi-stable characteristics. From the Figure 7, it can be observed that as $d_{g}$ increases, the system goes through bi-stable, tri-stable, and mono-stable in sequence. As shown in Figure 7a, when $d_{g}=6\;\mathrm{mm}$, bi-stable motion can be achieved. As shown in Figure 7b, when $d_{g}=8\;\mathrm{mm}$, the system can easily cross the potential barrier and move back and forth between the three steady-state positions, showing a tri-stable characteristic. At this time, the displacement and speed of the end of the beam are greater than other states, and the response displacement reaches 18 mm. As $d_{g}$ continues to increase, the force between the magnets is small, the system potential energy has only one potential well, and there is no multistable characteristic. Under low-level excitation, the system response displacement and velocity are both small.

#### 3.2.3. The Influence of Excitation Intensity A

When d=15 mm, $d_{g}=8\;\mathrm{mm}$, Ω=0.5, the excitation acceleration a is changed. Figure 8 shows the displacement-velocity phase diagrams with accelerations of $5\;m/s^{2}$, $7\;m/s^{2}$ and $12\;m/s^{2}$.

From Figure 8, we can see that: under the above conditions, the system potential energy curve has three potential wells. When the excitation is low, the system cannot cross the potential barriers on both sides and can only make periodic motions in the middle potential well. When $A=5\;m/s^{2}$, the system moves in a mono-stable state, with a small periodic vibration centered on the intermediate balance point. Increasing the excitation acceleration, the energy obtained by the system increases. When $A=7\;m/s^{2}$, the displacement of the beam increases, but the excitation magnitude is still not enough to make the system cross the two barriers. When the excitation acceleration increases to $12\;m/s^{2}$, the system presents a tri-stable motion characteristic, moving periodically between the three potential wells. Combining the previous analysis of the influence of the variable magnetic distance on the system dynamics, when the system presents a bi-stable or tri-stable state of motion, the steady-state points on both sides of the displacement-velocity phase diagram are asymmetrical. The speed on one side is higher than the other side, which is caused by the asymmetry of the restoring force of the linear-arch beam.

## 4. Experimental Validation

In order to verify the correctness of the theoretical analysis of the linear-arch beam tri-stable piezoelectric energy harvester, an experimental platform was built for experimental verification. The experimental device is shown in Figure 9: the linear-arched beam is fixed on the base by a clamp and is perpendicular to the horizontal sliding table of the vibrating table. Two external magnets B and C are fixed on the base, mutually exclusive with the magnet A. In the experiment, the excitation signal is set by the computer, and the sinusoidal signal is sent out by the vibration controller (VT-9008), which is amplified by the power amplifier (GF-20) and output to the vibration table (E-JZK-5T). The vibration table operates according to the preset excitation signal. Through the reflective sticker on the top of the arched section beam, the laser vibrometer (LV-S01, resolution: 1 um/s) can measure the speed of the cantilever, by using a handheld vibrometer (coco80, sampling rate: 2 KHz) to collect data.

Figure 10 is the experimental results of displacement-velocity of linear-arch composite beams with different excitation accelerations when d=15 mm,$\;d_{g}=8\;\mathrm{mm}$ and f=8 Hz. As shown in Figure 10a, when $A=5\;m/s^{2}$, the end displacement is 2 mm. As the excitation acceleration increases, the displacement amplitude increases. When $A=7\;m/s^{2}$, the end displacement increases to 2.5 mm. It shows that when the excitation is low, the system cannot cross the barriers on both sides and can only move in the well. Figure 10c is the displacement-velocity phase diagram at $12\;m/s^{2}$. The system can cross the potential barrier and show the characteristics of tri-stable motion. As shown in Figure 11, the three stable positions of the tri-stable system can be clearly seen.

Figure 12 shows the experimental results of displacement-velocity of linear-arched beam under different magnetic distances when f=8 Hz and $A=12\;m/s^{2}$. As shown in Figure 12a, when $d=22\;\mathrm{mm},d_{g}=8\;\mathrm{mm}$, the system has only one potential energy well, and the system moves periodically in the well. As shown in Figure 12b, when $d=16\;\mathrm{mm},d_{g}=6\;\mathrm{mm}$, the system has two potential wells, and the excitation intensity is sufficient for enabling the system to cross the potential barrier to achieve bi-stable motion. Figure 13 shows the two stable positions of the system. When $\;d=16\;mm,d_{g}=8\mathrm{mm}$, as shown in Figure 12c, the system cyclically moves between three stable positions, showing three-stable characteristics, and the displacement amplitude is greatly increased.

Comparing the experimental and simulation results, it can be seen that when the system response displacement is small, such as mono-stable motion, or the excitation is low, the experimental results are in good agreement with the simulation results, and the displacement amplitude error is about 8%. When the excitation is large or the system is in bi-stable and tri-stable motion, the experimental results shown in Figure 10c and Figure 12b,c have slightly obvious errors, and the experimental phase diagram appears to be tilted. The main reasons are: (1) there are processing errors in the production of the linear-arch composite beam TPEH; (2) the phase diagram obtained by the experiment is asymmetric and inclined, while the phase diagram obtained by the simulation is not inclined and symmetrical. This is because the gravity factor is not considered in the simulation; (3) due to the existence of the arched part in the beam, there is a deviation in the data collected by the laser vibrometer.

## 5. Conclusions

In this paper, a dynamic model is established for the linear-arch composite beam TPEH. The dynamic equation is numerically solved by the fourth-order Runge–Kutta algorithm, and the influence of magnet spacing and excitation intensity on system dynamics is analyzed. The influence of characteristics is verified by experiments to verify the correctness of the theoretical analysis. The following main conclusions are obtained from simulation and experiment:

(1) When keeping dg unchanged, by changing d, the system can form a mono-stable system and a tri-stable system. When keeping d unchanged, increasing $d_{g}$, the system forms a bi-stable, tri-stable and mono-stable system in sequence. When the system moves in a tri-stable state, the vibration response displacement of the system increases significantly.

(2) Take the energy harvester with $d_{g}=8\;\mathrm{mm}$ and d=16 mm as an example. Under this condition, the potential energy curve of the system has three potential wells, the depth of the potential wells is relatively shallow, and the width is relatively uniform; this helps the system to achieve a large response under low excitation.

(3) As the excitation level increases, it is easier for the system to cross the barrier to achieve inter-well movement, and the response displacement of the energy harvester increases.

(4) The asymmetric restoring force of the linear-arch beam results in an asymmetric potential well in the potential energy curve, which provides a new solution for the application of energy harvester in a low excitation environment.

## Figures and Tables

**Figure 1 micromachines-12-01045-f001:**
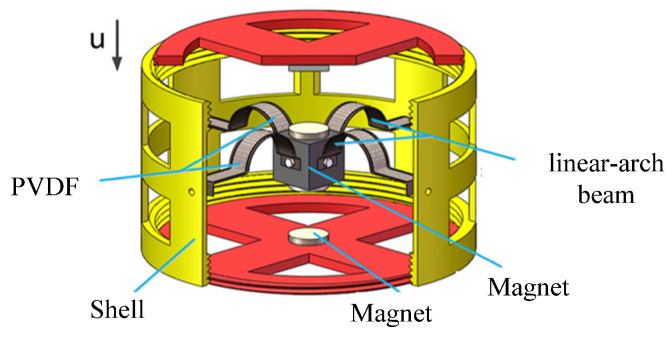
Multifield coupled multidirectional piezoelectric energy harvester.

**Figure 2 micromachines-12-01045-f002:**
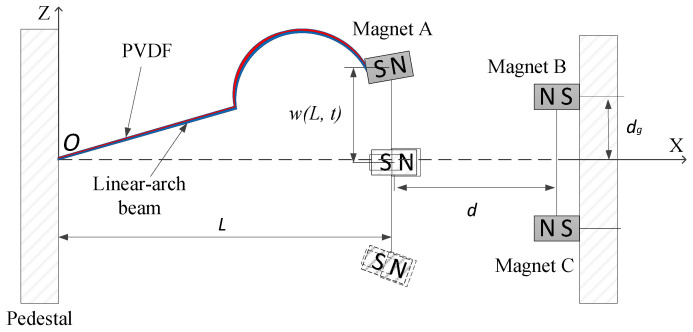
Schematic diagram of linear-arch beam TPEH.

**Figure 3 micromachines-12-01045-f003:**
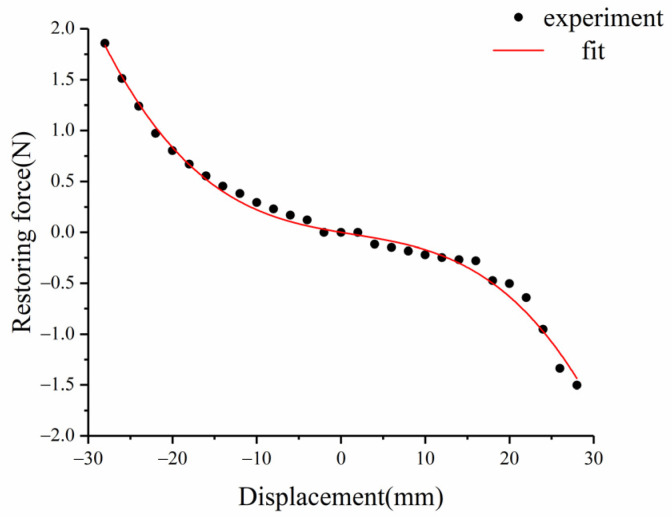
Nonlinear restoring force of linear-arch beam.

**Figure 4 micromachines-12-01045-f004:**
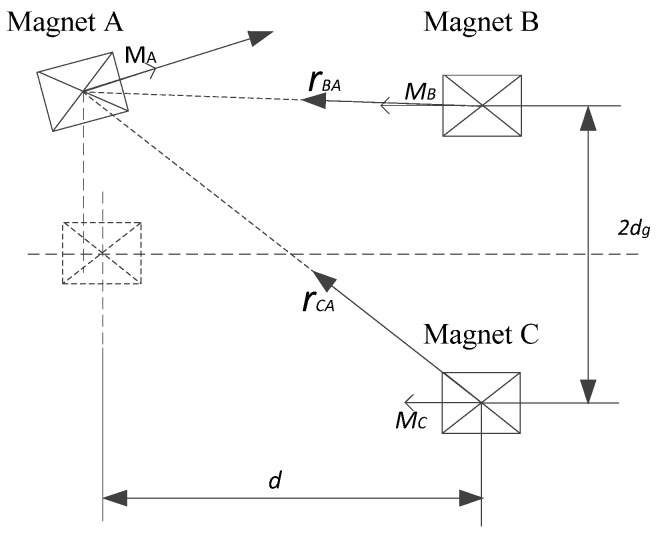
Nonlinear magnetic force model.

**Figure 5 micromachines-12-01045-f005:**
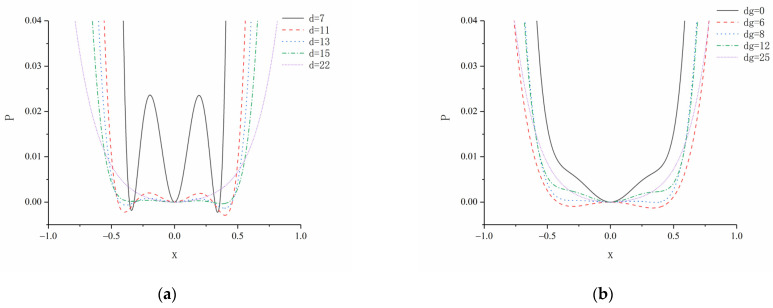
System potential energy of different magnetic distance: (**a**) change the horizontal magnetic distance d; (**b**) change the vertical magnetic distance $d_{g}$.

**Figure 6 micromachines-12-01045-f006:**
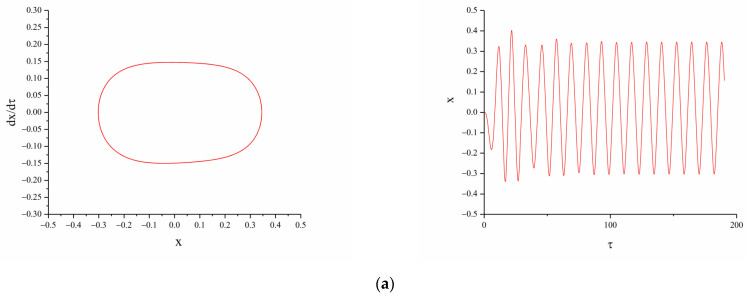
Phase portrait and time-displacement diagram of different magnetic distance d: (**a**)  d=22 mm; (**b**)  d=15 mm; (**c**)  d=13 mm.

**Figure 7 micromachines-12-01045-f007:**
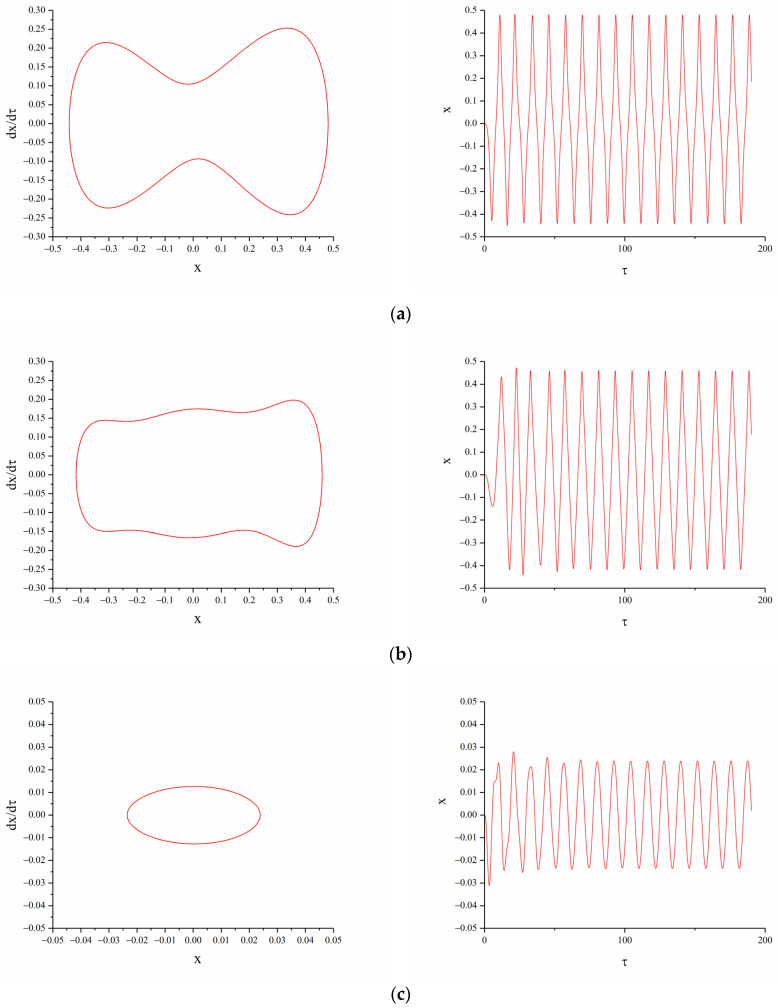
Phase portrait and time-displacement diagram of different magnetic distance $d_{g}$: (**a**) $\;d_{g}=6\;\mathrm{mm}$; (**b**) $\;d_{g}=8\;\mathrm{mm}$; (**c**) $\;d_{g}=12\;\mathrm{mm}$.

**Figure 8 micromachines-12-01045-f008:**
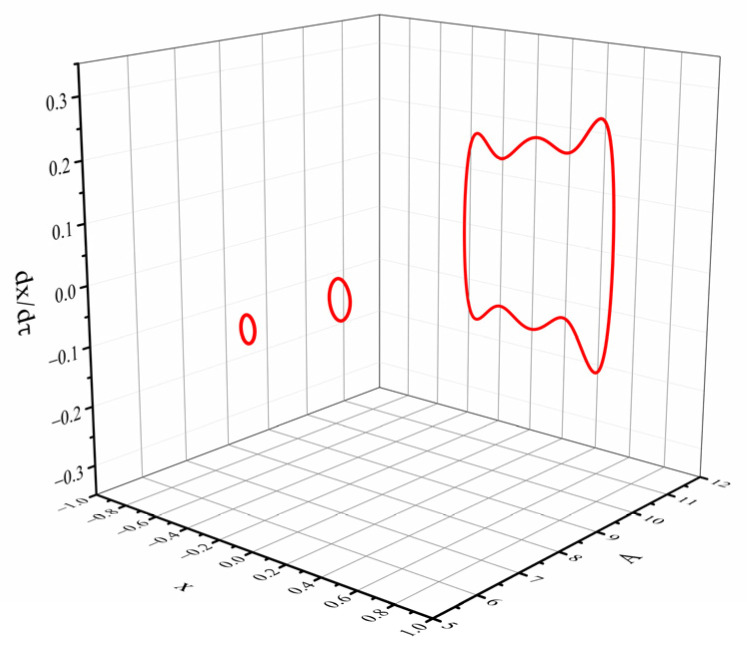
Phase portrait of different excitation acceleration A.

**Figure 9 micromachines-12-01045-f009:**
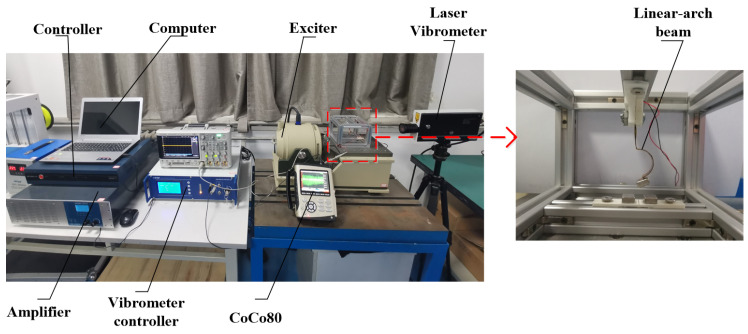
Experimental platform and TPEH.

**Figure 10 micromachines-12-01045-f010:**
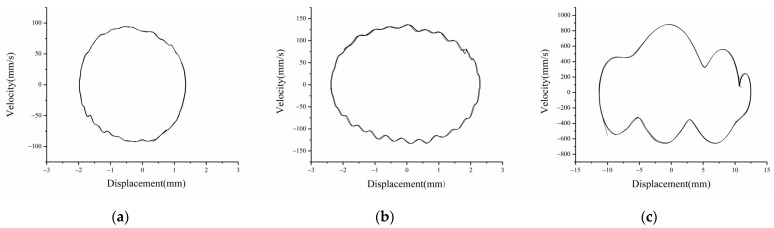
Experimental phase portrait of different excitation accelerations: (**a**) $\;a=5\;m/s^{2}$; (**b**) $\;a=7\;m/s^{2}$; (**c**) $\;a=12\;m/s^{2}$.

**Figure 11 micromachines-12-01045-f011:**
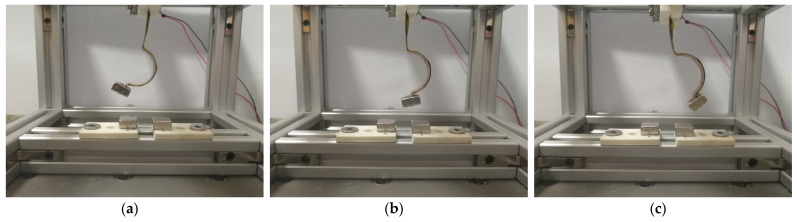
Three stable equilibrium positions: (**a**) stable1; (**b**) stable2; (**c**) stable3.

**Figure 12 micromachines-12-01045-f012:**
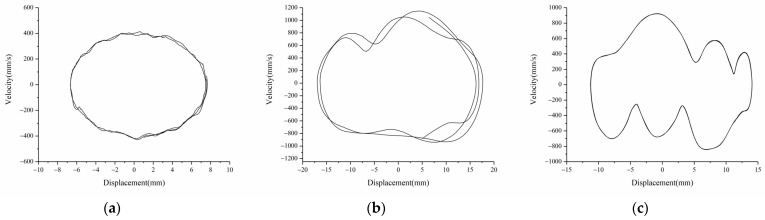
Experimental phase portrait of different magnetic distance: (**a**) $\;d=22\;mm,d_{g}=8\;\mathrm{mm}$; (**b**) $d=16\;\mathrm{mm},d_{g}=6\;\mathrm{mm}$; (**c**) $d=16\;\mathrm{mm},d_{g}=8\;\mathrm{mm}$.

**Figure 13 micromachines-12-01045-f013:**
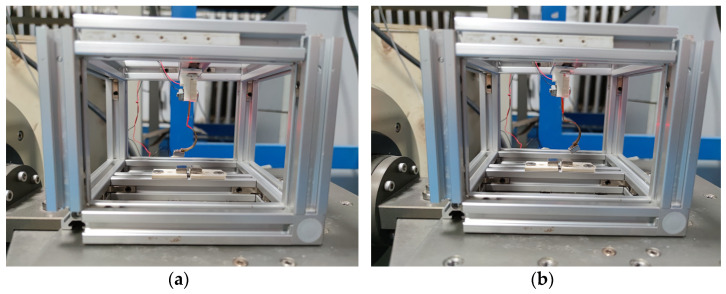
Two stable equilibrium positions: (**a**) stable1; (**b**) stable2.

**Table 1 micromachines-12-01045-t001:** Structure and material parameters of TPEH.

Parameter	Value	Parameter	Value
linear-arch beam		Piezoelectric layer	
$L_{1}*w*h_{s}\left({\mathrm{mm}}^{3}\right)$	20∗8∗0.2	$L_{2}*w*h_{s}\left({\mathrm{mm}}^{3}\right)$	40∗8∗0.2
r (mm)	10	Permittivity constant(F/m)	$110\times 10^{-12}$
$\mathrm{Young}’s\;\mathrm{modulus}\;(N/m^{2})$	$128\times 10^{9}$	$\mathrm{Young}’s\;\mathrm{modulus}\;(N/m^{2})$	$3\times 10^{9}$
$\mathrm{Density}\;(\mathrm{kg}/m^{3})$	8300	$\mathrm{Density}\;(\mathrm{kg}/m^{3})$	1780

## Data Availability

Not applicable.

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
