# Peer review of "Time-Domain Dynamic Characteristics Analysis and Experimental Research of Tri-Stable Piezoelectric Energy Harvester"

_micromachines, 2021, doi:10.3390/mi12091045_

Round 1
Reviewer 1 Report
This paper presents time-domain dynamic characteristics and experimental research of tri-stable piezoelectric energy harvester. The topic is interesting and the authors’ writing is clear. Below are some suggestions which might improve the quality of the paper:
- It would be best if the authors can nondimensionalize the analysis. For example, the authors present the conditions ? = 12?⁄?2,?? = 8??,? = 9H? in Figure 5. The specific numbers make sense in this research, however, it doesn’t mean anything to the readers. I give an example, if you read a paper stating the damping coefficient is 100 N/(m/s) or 0.01 N/(m/s), did you immediately know what that means? I don’t think anybody knows what is that. However, if the authors present the camping ratio zeta=0.1, or 0.5, you will immediately know the implications, as long as you take any vibration courses. So I would suggest using nondimensionalized values somehow. If you can nondimensionlize ?1 = 125 N/m,?2 = 254.586 N/m2,?3 = −56681.2 N/m3, another researcher can compare their result with your result and have more insights. Even the other research might be in the MEMS scale, they can still compare theirs with yours.
- Figure 5-7 are just some examples of vibration, which I will not call it analysis. Many other papers have that. It would help if you can nondimensionalize everything and figure out the critical parameter points between intra-well and inter-well behavior, maybe even more insights. The purpose of the paper is to serve the readers, can you do some more in-depth analysis to let the readers appreciate your work. Now it is just a few examples, and imagine you are the reader, how much information and insight can you obtain?
- Another thing is that the sampling rate in the time domain in Figures 5 and 6 is too low.
- Why uses a linear-arch beam? Does it have a better performance compared with other shapes? Why not a straight beam?
- Usually, in the last paragraph of the introduction section, the authors will indicate what are the rest section all about.
- “Aiming at the power supply requirements of wireless monitoring nodes in underground coal mines, in order to adapt to the low-frequency and random characteristics of excavation excitation, this paper designs a linear-arch beam TPEH based on the previous research of the research group.” — “underground coal mines” is a specific application, it might be important for a project report, but it is not important here in this paper. This design is not limited to this application. I think other tri-stable papers also address “low-frequency and random characteristics”, what is your contribution?
- I can see the authors put tremendous effort and I enjoy the writing. However, the authors should read more on how to write a paper (there are many books that you can refer to). For example, usually, the introduction part consists of three steps: Establishing a Territory [the situation], Establishing a Niche [the problem], Occupying the Niche [the solution]. “Obtaining energy from the environment to supply power to wireless sensor nodes is expected to solve this problem.” — I would rather say that “Obtaining energy from the environment has the potential to solve this problem and address the power supply issue.” Then this topic becomes interesting. “Compared with other nonlinear methods, it is relatively easy to construct a nonlinear piezoelectric energy harvester by introducing a magnetic field.” — I would expect you to state your contribution here. However, you indicate you do this because it is easy. This is not acceptable. After you have introduced all the literature, you should indicate a gap: there are many researches, however, there is still some problem that has not been solved. Then my research addresses the problem (fill the gap). In this way of writing, the review can see your scientific contribution, and your readers will appreciate your work. In the current format, I can only see you present your work, I did not see your scientific contribution.
- For the conclusions, I think everybody knows Point 1. It might not be the conclusion of this paper. Also, adjusting the distance is but just one method to make it work. There are also other methods, such as tuning the magnetic field. Also, Point 2 is true for all tri-stable systems, and it is not your finding. What is your contribution?
Hope this can help to improve the paper quality.
Reviewer 2 Report
The paper presented a dynamic analysis and experimental study of TPEH. The paper is well organized and written. The reviewer only has some question for the experimental study:
1) The reviewer did not see a results comparison between the experimental study and the theoretical model.
2) The experiment only tested different excitation acceleration. What about different horizontal and vertical distances?
3) What about the performance of the design system at different excitation frequencies?
4) The experiment section should provide more details such as the vibrometer model number, sensitivity, the sampling rate of data acquisition, etc.
Round 2
Reviewer 1 Report
Sorry that the comment might be harsh. My goal is to make this paper in a very good shape and appreciated by the readers.
Reviewer 2 Report
The reviewer has no further comments.